# Applications of Delayed Luminescence for tomato fruit quality assessment across varied Sicilian cultivation zones

Salvina Panebianco[1]*, Eduard van Wijk[2], Yu Yan[2], Gabriella Cirvilleri[1], Alberto Continella[1], Giulia Modica[1], Agatino Musumarra[3,4], Maria Grazia Pellegriti[4], Agata Scordino[3,5]

1 Dipartimento di Agricoltura, Alimentazione e Ambiente, Università di Catania, Catania, Italy, 2 Department of Biophotonics, Meluna Research, Wageningen, Netherlands, 3 Dipartimento di Fisica e Astronomia, Università di Catania, Catania, Italy, 4 Istituto Nazionale di Fisica Nucleare – Sezione di Catania, Catania, Italy, 5 Istituto Nazionale di Fisica Nucleare – Laboratori Nazionali del Sud, Catania, Italy

* salvina.panebianco@phd.unict.it

**Data Availability Statement:** All relevant data are within the paper and its Supporting information files.

## Abstract

The food industry places significant emphasis on ensuring quality and traceability as key components of a healthy diet. To cater to consumer demands, researchers have prioritized the development of analytical techniques that can rapidly and non-invasively provide data on quality parameters. In this study, we propose to use the Delayed Luminescence (DL), an ultra-weak and photo-induced emission of optical photons, as a tool for a rapid evaluation of quality profile associated with fruit ripening, in support of traditional analysis methods. Delayed Luminescence measurements have been performed on cherry tomatoes, with and without the PGI "Pomodoro di Pachino" certification, harvested from two different growing areas of south-eastern Sicily (Italy). Then, DL emissions were correlated with soluble solid content and titratable acidity values, which are known to affect the flavor, the commerciality and the maturity degree of tomato fruits. In addition, we evaluated the changes in the DL parameters with respect to the geographical origin of the cherry tomatoes, with the aim of testing the possibility of applying the technique for identification purposes. The signals of Delayed Luminescence appeared to be good indicators of the macromolecular structure of the biological system, revealing structural changes related to the content of total soluble solids present in the juice of tomatoes analyzed, and they appeared unsuitable for authenticating vegetable crops, since the differences in the photon yields emitted by tomato Lots were not related to territory of origin. Thus, our results suggest that DL can be used as a nondestructive indicator of important parameters linked to tomato fruit quality.

## 1. Introduction

Consumers' request for high-quality foods with a certified geographical identity, like PDO (Protected Designation of Origin) and PGI (Protected Geographical Indication) products, has greatly increased in the last years, widening the fraud phenomenon due to the higher economic value of these products. As a result, analytical methods able to determine the quality

**Funding:** Funded by Incentive Plan for Research (PIA.CE.RI.) 2020-22 line 2 of the University of Catania, Research Projects "NaTI4Smart" and "MEDIT ECO", National Operational Programme (PON) on Research and Innovation 2014-2020 "POFACS (ARS01_00640)" - Italian Ministry of University and Research. The funders had no role in study design, data collection and analysis, decision to publish, or preparation of the manuscript.

**Competing interests:** The authors have declared that no competing interests exist.

and the geographical origin of products fast and easy has become increasingly necessary. In this context, the focus of this research was to authenticate and determine the quality of PGI "Pomodoro di Pachino" using non-destructive physical analysis methodologies. Results obtained by the XRF technique [1, 2] clearly showed that elemental composition of fruits was correlated to their territory of origin, which allows to identify typical elements or pollutants even when they appear in trace amounts. This clearly establishes a plethora of applications aimed at preventing food fraud and certifying food quality. These results boost the implementation of further non-destructive physical techniques.

Several analytical techniques, such as NIRs/FIR, Raman or Nuclear Magnetic Resonance (NMR) spectroscopy, are nowadays widely used in the agri-food industry to determine the degree of fruit ripeness, nutritional quality of food and/or to determine the presence of chemical residues that are dangerous to human health [3–8]. Some of these techniques have also been successfully used for detecting geographical origin of different vegetal crops, including the Sicilian PGI 'Pomodoro di Pachino' [9, 10]. Among the non-destructive physical techniques, Delayed Luminescence has also attracted particular interest in the last decades. It measures the optical photons released by a sample as a result of its excitation by light, long time after the switching off of the illumination source. The phenomenon of delayed light emission from biological systems (in few seconds) following exposure to illumination was observed for the first time in green plants in 1951 [11]. The amount of light emitted was $10^3$–$10^5$ times lower in intensity with respect to the fluorescence, a low-level signal whose lifetime spectrum spans from $10^{-7}$ to more than 10 s. This phenomenon has been associated with triplet- or metastable-state species exhibiting intrinsically long lifetimes.

Nowadays, we know that DL emission is not only associated with green plants, but it is an indicator of system biological state. This allowed to explore the potential of DL in different fields, from quality control to clinical investigations. In the agri-food sector, the technique has been used to assess the germination capacity of seeds [12, 13], photosynthetic activity in plant leaves and the ripening of fruits that lose chlorophyll as a result of ripening [3]. In this regard, a study carried out on tomato fruits at different stages of ripening has highlighted significant changes in the emissions of DL related to the color of fruits and to their respiration [14].

In the biological systems, DL phenomenon has been correlated to the dimension of ordered structures where it has been measured, exhibiting drastic changes when the structure was altered. A model that assumes that DL is connected with the formation and dissociation of non-linear coherent self-trapped (localized) electron states (solitons and electro-solitons) has been implemented and it succeeded in explaining a certain number of DL features [15–18]. For this reason, it is reasonable to infer that fruits grown in different farmlands, thus having different chemical composition, can give rise to different light signals, which can be used to draw information about quality parameters.

In fruit and vegetable products, quality depends on several physicochemical parameters (skin color, sugar and organic acid content, pH, moisture and ash content) that are not always easily detectable. These factors are closely related to the degree of fruit ripening and are often used to determine the best period for harvesting. Fruit ripening is a complex process that involves irreversible changes in color, texture, taste and chemical composition. Currently, the appropriate time for harvesting is mainly estimated by counting days after flowering or through color and texture visual inspections (subjective parameters). In addition, to define the best harvest time, sugar/acid ratio is often determined. For this purpose, traditional instruments (refractometer, pH meter) are used, albeit they have the disadvantage of destroying the analyzed matrix and requiring a large number of samples to reduce statistical variability.

In this context, the main objective of this study was to investigate the possibility of using DL technique as alternative to traditional analysis methods to determine, in a fast and non-

destructive way, the quality and/or degree of fruit ripening. To this end, we analyzed the correlations between DL parameters (obtained by the DL decay curves) and physicochemical parameters associated with fruit ripening (color, total soluble solid content, titratable acidity). Colorimetric, chemical and DL measurements were performed on four different Lots, consisting of cherry tomatoes from two different agricultural zones of south-eastern Sicily. In detail, two Lots came from greenhouses located in Ispica, within the Pachino district (PGI-site), and the other two from greenhouses in the province of Ragusa (non-PGI sites). When comparing Lots from different cultivation zones, we also considered the possibility of authenticating the PGI Pachino tomato by using DL technique and we evaluated possible correlations between DL parameters and the geographical origin of the samples.

## 2. Materials and methods

### 2.1 Fruit sampling

Cherry tomato fruits cv. Creativo were harvested in four different greenhouses located in south-eastern Sicily, from PGI and non-PGI sites, at the same time. In detail, the area of origin of the tomato Lots included two greenhouses located in Ispica municipality (PGI sites), within the Pachino district, (Lots 3 and 4) and two greenhouses located in Marina di Ragusa and Santa Croce Camerina municipalities (non-PGI sites), in the Ragusa province (Lots 1 and 2). A total of four Lots were collected, taking one Lot from each greenhouse (Fig 1). All Lots were kindly provided by farm owners.

In Ispica, the tomato cultivation was carried out following the rules indicated in the Pachino tomato production specifications, providing irrigation by groundwater from local wells, characterized by salinity, because of the nearby sea (electrical conductivity ranging from 1500 to 10000 μS/cm). The salinity of the irrigation water in Marina di Ragusa and Santa Croce Camerina is generally lower than in Ispica.

Considering the large size of the greenhouses, for each sampling, tomatoes were randomly harvested from different points and pooled in a single Lot. In this way, it was possible to obtain

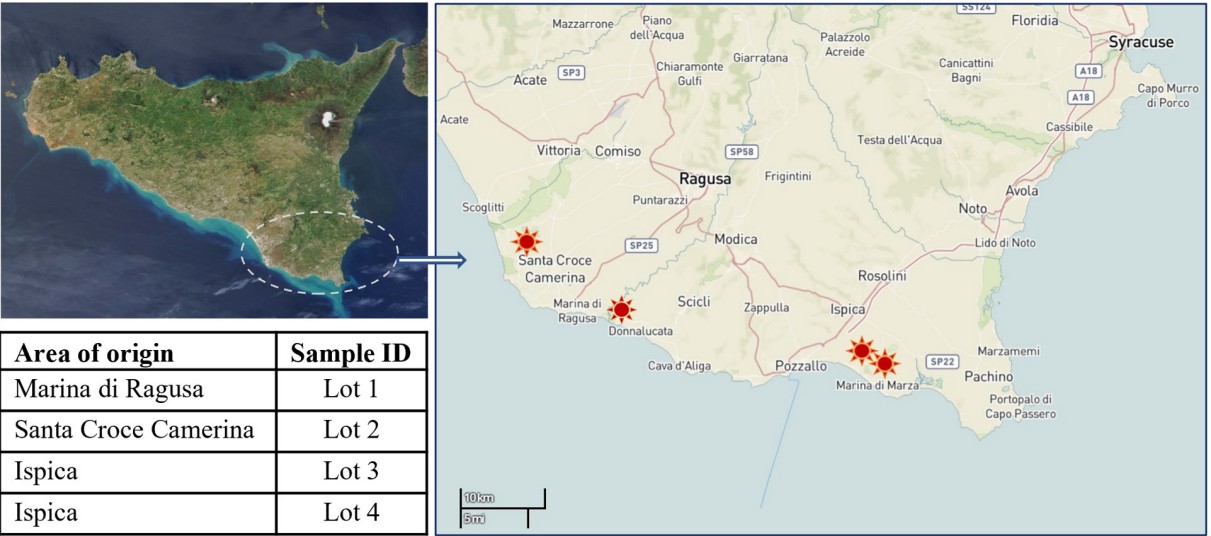

| Area of origin | Sample ID |
|---|---|
| Marina di Ragusa | Lot 1 |
| Santa Croce Camerina | Lot 2 |
| Ispica | Lot 3 |
| Ispica | Lot 4 |

**Fig 1. Area of origin of tomato Lots collected for this study.** Lots 3 and 4 were certified with the label Protected Geographical Indication (PGI). The imageries were freely downloaded from NASA Earth Observatory (http://earthobservatory.nasa.gov/) and USGS EROS (Earth Resources Observatory and Science–EROS–Center, http://eros.usgs.gov/#). The figure on the right side is similar but not identical to the original image and is therefore for illustrative purposes only.

a uniform and representative Lot from each greenhouse. Tomato fruits were aseptically removed from plants together with their bunches, put in plastic bags and transported to the laboratory.

In the framework of this research activity, we analyzed tomato samples at full ripeness stage (skin with intense red color). For each Lot, 35 ripe fruits were selected, for a total of 140 fruits constituting the whole experimental set. The selected fruits appeared similar to each other in color, size and weight (Fig 2). Part of these fruits were employed to assess the physicochemical parameters concerning tomato quality, while the rest were subjected to DL measurements, as specified below. Throughout the period of measurements after harvesting, tomatoes were kept at room temperature and in dark conditions.

## 2.2 Physicochemical parameters

For each Lot, the size and weight measurements were performed on 20 tomato fruits and the same fruits were used for DL measurements. Fruit weights were determined using a digital balance (Diamond, model 500 with an accuracy of 0.1 g). Fruit sizes (equatorial diameters) were measured with an electronic digital caliper (Digital Caliper 150 mm, accuracy of 0.01 mm); this measurement was repeated twice.

Due to the destructive nature of chemical measurements, a different set of 15 fruits were selected for each Lot and used for chromatic measurements. The chromatic data was obtained by using a portable colorimeter model PCE-XXM 30 (PCE-Deutschland GmbH, Meschede, Germany) with a repeatability of $\Delta E^*_{ab} \leq 0.1$. A lamp with a wavelength of 400–700 nm (LED lamp) was built into the colorimeter as a light source. The measuring spot of the colorimeter

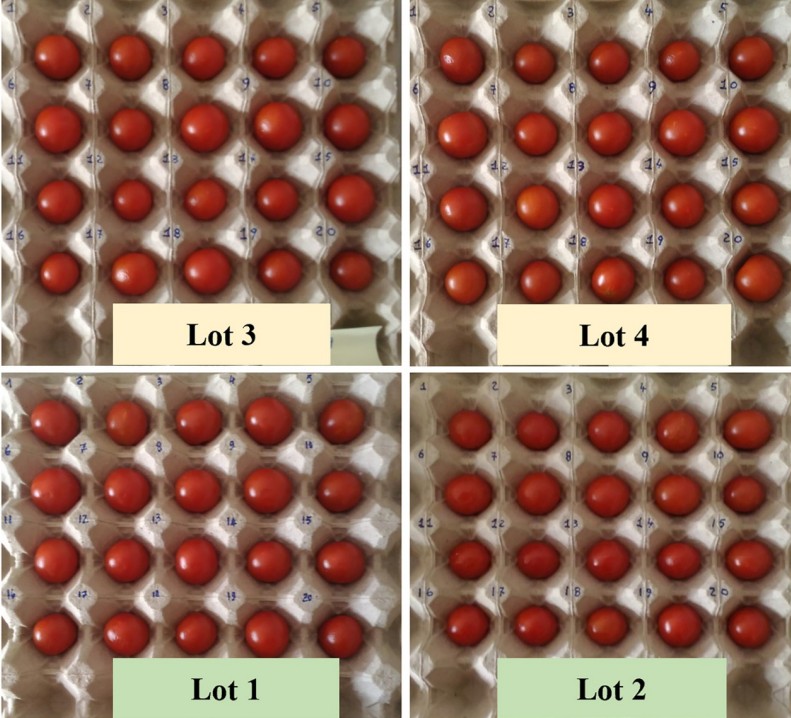

**Fig 2. Cherry tomato fruits selected for DL measurements.** The Lots 1 and 2 were collected from non-PGI sites, whereas the Lots 3 and 4 from farms in the Pachino district, recognized by the European Community with the label PGI (see Fig 1).

was 8 mm in size. Fruit chromaticity was recorded according to the $L^*$ $a^*$ and $b^*$ color space coordinates defined by the International Commission on Illumination (CIELAB). For each Lot, $L^*$ $a^*$ and $b^*$ chromatic values were determined on the 15 fruits, with measurements at two different points on each fruit (for a total of 30 measures per Lot). The $L^*$ value determines the amount of light reflected. The $a^*$ value indicates the chromaticity on the red (+ values) or green (− values) axis, and the $b^*$ value indicates the chromaticity of the yellow (+ values) or blue (− values) axis [19].

From $a^*$ and $b^*$ values we calculated $a^*/b^*$ color ratio, chroma ($C^*$) and hue angle ($H^°$) on the basis of equations reported by [20]:

- $chroma = \sqrt{(a^{*2} + b^{*2}}$;

- $H^° = \tan^{-1}(b^*/a^*)$ (when $a^* > 0$ and $b^* \geq 0$) or

- $H^° = 180 + \tan^{-1}(b^*/a^*)$ (when $a^* < 0$).

Concerning the chemical measurements, the fruits were crushed in order to determine pH, total soluble solid (TSS) content, expressed as °Brix, and titratable acidity (TA) expressed as g/L of citric acid equivalent. Maturity index (MI) was calculated as the ratio between TSS and TA. For each Lot, the juice of five fruits per three replicates was extracted using a commercial juice extractor. The TSS content was determined using a digital refractometer (Atago CO., LTD, model PR-32 α, Tokyo, Japan) with a °Brix detection range of 0% to 32%, whereas pH and TA of the tomato juices were determined using an automatic titration device (Hach, Titra-Lab AT1000 Series, Lainate, Milano) with 0.1 N NaOH up to pH 8.1.

All experimental data were analyzed through the one-way analysis of variance (ANOVA StatSoft Inc, Tulsa, OK, USA) at an error level of 0.05 and compared by means of Fisher's test to establish possible significant differences among the various parameters.

## 2.3 DL measurements

The apparatus used for measuring the delayed light emission was provided by Meluna Research (Wageningen, The Netherlands) and described in [21, 22]. It included a black chamber with a sample holder suitable for a 10 cm Petri dish. The dark chamber was connected to a photomultiplier tube installed in a vertical position (PMT, model 9558QB; Electron Tubes Enterprises Ltd., Ruislip, UK). The PMT, enhanced for single photon counting, was sensitive in the 300–800 nm wavelength range. Moreover, it was cooled to -25°C to reduce the PMT thermal noise. The samples were illuminated for 10 s by using a white tungsten halogen lamp with an emission spectrum ranging from 350 to 950 nm (20 W, model 284–2812, Philips, Germany). The counts were recovered by a time binning of 100 ms, during 30 s after excitation, thus resulting in DL spectra of 300 data points. The amplification of the photon emission signal was carried out by using a fast spectroscopy preamplifier (model 9301 ORTEC, Oak Ridge, TN) connected to the PMT, whereas data acquisition was performed by using a photon counting card (C9744; Hamamatsu, Iwata, Japan) installed in a personal computer. One measurement was performed on each fruit.

# 3. Results

## 3.1 Physicochemical parameters

Table 1 reports size, weight and chromatic parameters as defined by the CIELAB system ($L^*$ $a^*$ and $b^*$ coordinates, $C^*$ and $H^°$ measurements, $a^*/b^*$ color ratio) obtained from the PGI and non-PGI tomato Lots. Equatorial size and weight values were the same in most tomato fruits

**Table 1. Equatorial size, weight and chromatic parameters (CIELAB system coordinates: $L^*$, brightness; $a^*$, red-green component; $b^*$, yellow-blue component; $C^*$, chroma; $H^\circ$, hue angle; $a^*/b^*$, color ratio) evaluated in tomato fruits coming from two different Sicilian growing areas.**

|  | Lot 1 | Lot 2 | Lot 3 | Lot 4 |
|---|---|---|---|---|
| Equatorial size (mm) | 27.9 ± 1.39 [a] | 26.7 ± 1.31 [b] | 26.9 ± 1.92 [b] | 26.3 ± 1.49 [b] |
| Weight (g) | 13.4 ± 1.6 [a] | 11.5 ± 1.4 [b] | 11.7 ± 2.2 [b] | 11.7 ± 1.7 [b] |
| $L^*$ | 32.4 ± 1.01 [c] | 33.3 ± 0.85 [b] | 34.0 ± 1.45 [a] | 33.9 ± 1.53 [ab] |
| $a^*$ | 12.8 ± 1.75 [b] | 14.5 ± 1.47 [a] | 14.2 ± 1.61 [a] | 14.2 ± 1.85 [a] |
| $b^*$ | 15.8 ± 1.08 [c] | 17.5 ± 1.52 [b] | 18.8 ± 1.84 [a] | 19.0 ± 2.31 [a] |
| $C^*$ | 20.4 ± 1.68 [c] | 22.4 ± 2.19 [b] | 21.4 ± 2.05 [bc] | 23.6 ± 2.29 [a] |
| $H^\circ$ | 51.2 ± 3.15 [c] | 50.6 ± 2.57 [c] | 57.9 ± 3.98 [a] | 53.0 ± 2.21 [b] |
| $a^*/b^*$ | 0.81 ± 0.09 [a] | 0.83 ± 0.07 [a] | 0.76 ± 0.06 [b] | 0.75 ± 0.09 [b] |

The size values represent the mean of 40 measures per Lot ± standard deviation (two measurements per 20 fruits); weight values represent the mean of 20 measures per Lot ± standard deviation; chromatic parameters, for each Lot, represent the mean of 30 measures ± standard deviation (two measurements per 15 fruits). Lots 1 and 2 were harvested in non-PGI sites, whereas Lots 3 and 4 in Pachino district (PGI-sites). Values followed by different letters within each row are significantly different at α ≤ 0.05 (Fisher's least significant difference test).

analyzed (only the Lot 1 was slightly larger than the others). These values ranged between 11.5–13.4 mm and 26.3–27.9 g, respectively.

Regarding the colorimetric measurements, it is worth noting that fruits of each Lot were selected by visual inspection. To test their color uniformity, the color difference $\Delta E^*_{ab}$ (in the CIELAB color system) between fruits of the same Lot was evaluated. The average value and its standard deviation were 2.8 ± 0.9 for Lot 1, 2.9 ± 0.6 for Lot 2, 3.5 ± 1.0 for Lot 3 and 4.2 ± 1.0 for Lot 4. All the values were close to the just noticeable difference. Contrarily, differences in color parameters were observed between the different Lots analyzed. In particular, the values of red—green ($a^*$) and yellow—blue ($b^*$) components, as well as the chroma and hue angle ($H^\circ$) values, were slightly larger for PGI certified tomatoes from Ispica (Lots 3 and 4), whereas the highest values of the $a^*/b^*$ ratio were found for non-PGI tomato Lots from Ragusa province (Lots 1 and 2) (Table 1).

With regards to the chemical parameters, TSS content was significantly different in the analyzed tomato Lots and did not depend on the PGI quality certification of fruits (Table 2). In detail, TSS content was the highest in Lots 1 and 4. Contrarily, no significant differences in the pH, citric acid and maturity index were detected among the Lots.

### 3.2 DL measurements

The ultraweak delayed photon emission (Delayed Luminescence, DL) was measured on tomato fruits following light irradiation. The analyzed cherry tomato fruits belonged to four

**Table 2. Chemical parameters evaluated in tomato fruits coming from two different Sicilian growing areas.**

|  | Lot 1 | Lot 2 | Lot 3 | Lot 4 |
|---|---|---|---|---|
| TSS (˚Brix) | 8.1 ± 0.2 [b] | 7.1 ± 0.3 [c] | 7.2 ± 0.3 [c] | 8.6 ± 0.3 [a] |
| pH | 4.80 ± 0.12 [a] | 4.57 ± 0.13 [a] | 4.83 ± 0.09 [a] | 4.57 ± 0.20 [a] |
| Citric acid (g/L) | 5.30 ± 0.22 [a] | 5.29 ± 0.74 [a] | 4.35 ± 1.32 [a] | 5.55 ± 0.14 [a] |
| Maturity index (TSS/TA) | 1.52 ± 0.08 [a] | 1.35 ± 0.21 [a] | 1.74 ± 0.45 [a] | 1.55 ± 0.03 [a] |

For each Lot, values represent the mean of 3 measures ± standard deviation. For each Lot, measurements were carried out on 3 replicates consisting of the juice of 5 tomato fruit). Lots 1 and 2 were harvested in non-PGI sites, whereas Lots 3 and 4 in Pachino district (PGI-sites). Values followed by different letters within each row are significantly different at α ≤ 0.05 (Fisher's least significant difference test).

different Lots, each representing the four greenhouses reported in paragraph 2.1. In particular, for each Lot, DL measurements were performed on 20 fruits (for a total of 80 measurements), characterized by similar color, size and weight (S1 Appendix).

The photon count rate, recorded over time after the light was turned off, was plotted to evaluate the trend of decay curves. The decay curves recorded for each Lot were then superimposed in order to study the internal variability of the four Lots. The DL decay curves obtained for the 20 tomato fruits belonging to the four tomato Lots are shown in Fig 3.

Excluding Lot 3, total DL photon yield emitted by tomato fruits ranged within one order of magnitude. The decay trends of Lots 1, 2 and 4 were found quite homogeneous, as they were obtained by tomatoes with similar features. The DL photon yield variability within Lot 3 was

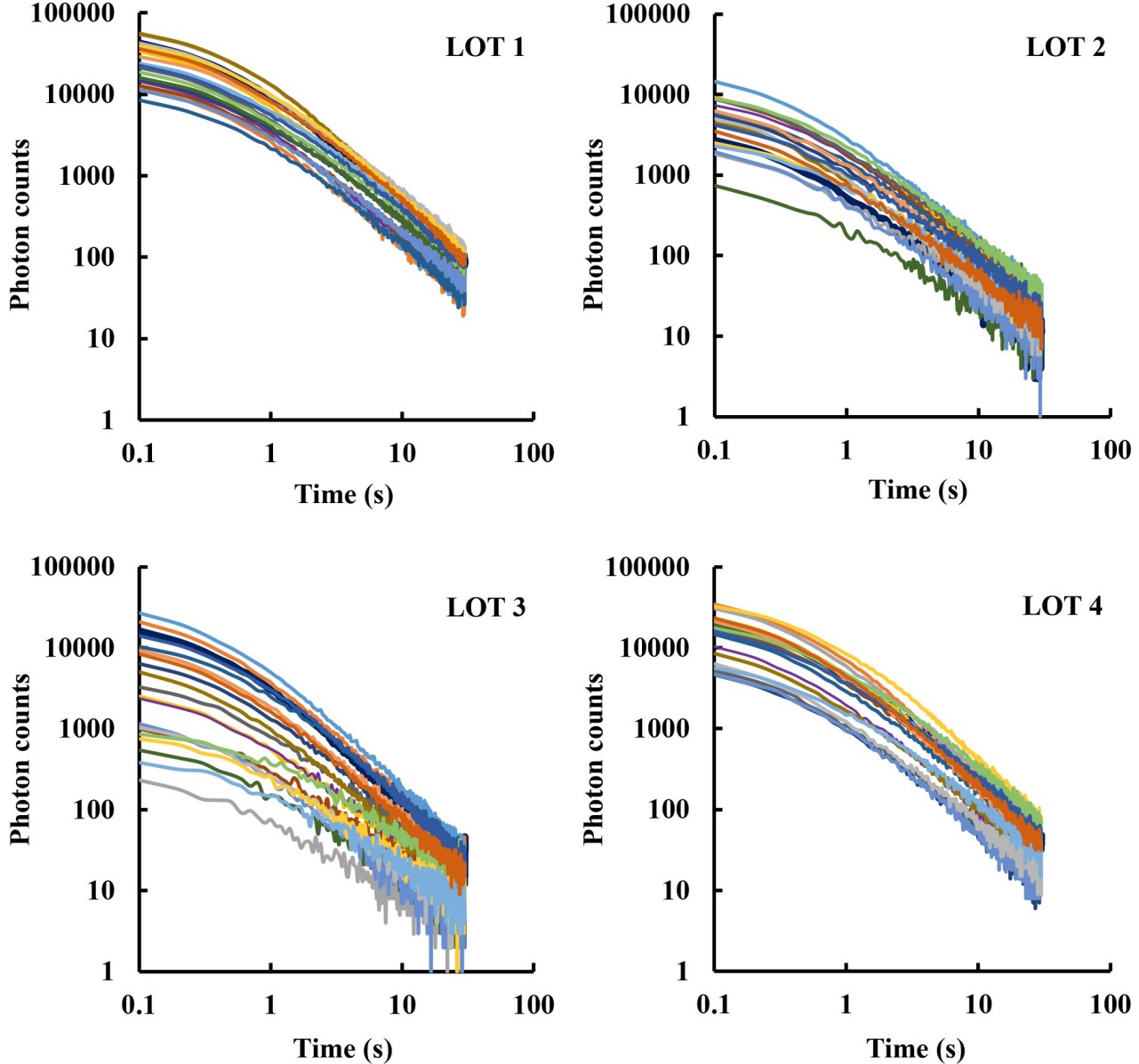

**Fig 3. DL decay curves of single fruits per lot.** DL measurements have been performed on 20 fruits per Lot, identified with different colors. Data are plotted on a log-log scale.

instead significantly higher than that observed for the other Lots. Colorimetric data do not support such variability, because there is no evidence of differences in color ($\Delta E^*_{ab}$) between fruits of Lot 3 with respect to the other Lots. Probably it could be related to the large spread in Maturity Index, but no definitive conclusion can be drawn at this moment.

By comparing the decay curves obtained from tomatoes belonging to different Lots, it was possible to unveil differences in the emission rate. In detail, the intensity of DL signals at t = 0.1s (first point of the decay curve) was larger for Lots 1 and 4 and lower for Lots 2 and 3, suggesting that differences between Lots could be related to their overall composition/structure rather than their geographical origin. To quantify the differences in trends of delayed light emitted by different Lots, thus simplifying their comparisons, experimental data were fitted using the following empirical decay function introduced by E. Becquerel in 1867:

$$I(t) = \frac{I_0}{\left(1 + \frac{t}{t_0}\right)^m} \tag{1}$$

This function has been largely used to describe DL decays obtained by a great variety of biological systems [23, 24] and it has already been used to characterize relaxation of other complex systems [25–28]. Indeed, relaxation of complex systems from non-equilibrium state towards equilibrium cannot be characterized by single rate coefficient but more appropriately a distribution of relaxation kinetics exists, which in turn can be related to the hierarchically organized structure of the energy landscape [25]. Fit parameters were evaluated according to the non-linear least-squares procedure described in [23].

Fig 4 shows the parameters $m$ and $t_0$ of Eq (1) obtained after normalization of data to the first experimental point, i.e. the photon rate $I_1$ at t = 0.1s. The $I_1$ values as a function of the fit parameter $m$ are reported in Fig 5.

In addition, we have developed a fast fitting procedure to obtain DL parameters in the shortest possible time. For this purpose, we tried to obtain a single representative trend for each Lot by using all the normalized measurements of the Lot, and evaluating, at each time instant, the mean value of the corresponding 20 DL rates at that instant, thus getting an average normalized decay trend.

Fig 6 reports the average normalized decay trends obtained for the four Lots, as explained, along with the corresponding theoretical trend Eq (1) (solid line). The parameters $t_0$ and $m$ obtained through the fitting procedure applied on the averaged decay curves were reported in Fig 7 and compared with the Lot' mean value of the DL fit parameters obtained by averaging the full set of fruit data. It appears that percentage differences between the mean value and corresponding value of the averaged trend were less than 5% for all the Lots, except for Lot 3. These results suggest that fitting of average normalized DL decay curves was a valuable tool to use when a rapid analysis of large amounts of data is required.

The parameters $t_0$ and $m$ obtained through the fitting procedure of decay curves associated to PGI and non-PGI tomato Lots were also compared with each other in order to verify the possibility of using luminescence for identification purposes.

By comparing data reported in Figs 4, 5 and 7, it was possible to highlight significant differences between the Lots analyzed, especially in the slopes (parameter $m$) of the hyperbolic trends. Regarding the mean values of DL parameters obtained from decay curves of 20 fruits (see Fig 7, white bars), the maximum values for the slopes were found for Lots 1 and 4. Significant differences were also found for the parameter $t_0$ and experimental first point $I_1$. As previously observed for TSS measurements, these differences did not depend on the geographical origin of the Lots.

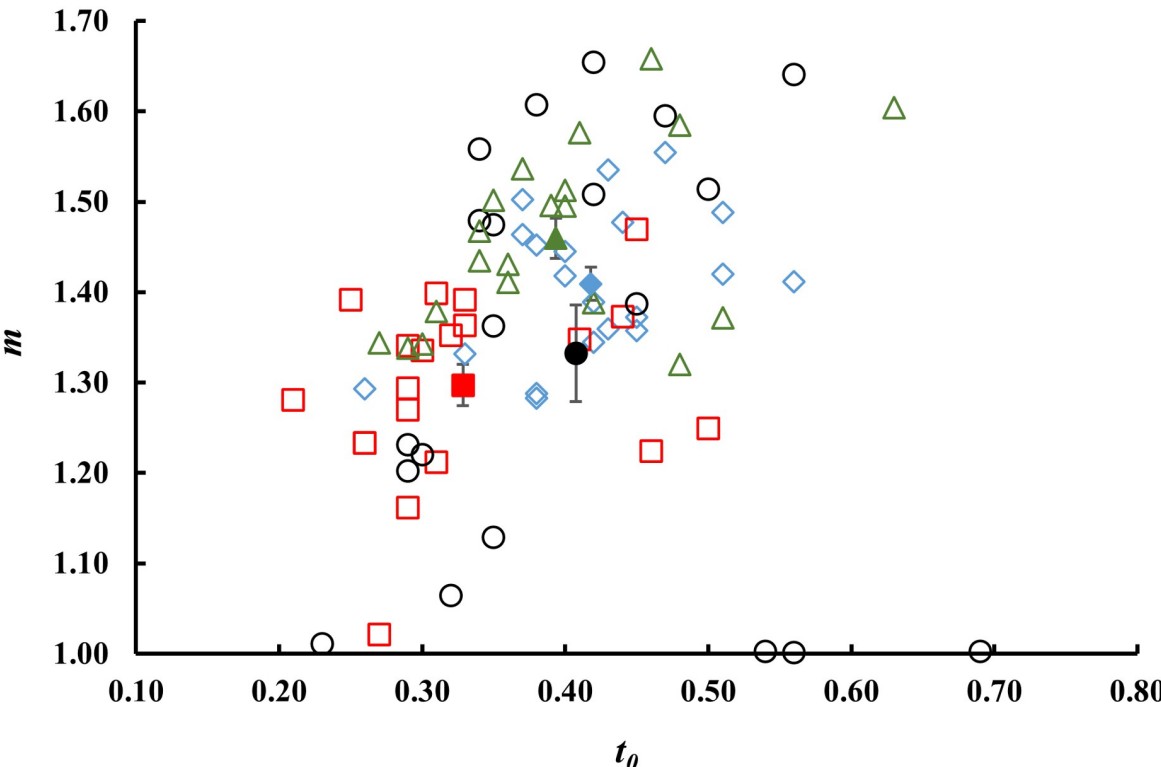

**Fig 4. Comparison between fit parameters.** DL parameters $m$ and $t_0$ obtained through the fitting procedure according to Eq (1) of the DL normalised decay curves in correspondence of each fruit which was part of a Lot: (diamond) Lot 1, (square) Lot 2, (circle) Lot 3, (triangle) Lot 4. Errors are within the marker size. Full markers represent mean values ±. S.E. for each Lot.

## 4. Discussion

Chromatic and chemical parameters determined in this study play an important role in the agri-food sector for quality management, since they are widely used to check quality standards during vegetable processing or to determine their harvesting time. Among chromatic parameters, the $a^*/b^*$ color ratio defined in the CIELAB system has been proposed by [29] as suitable for establishing the degree of tomato ripeness, depending on how much its value is close to one. More recently, other authors have highlighted positive correlations between the $a^*/b^*$ ratio and the lycopene content, whose biosynthesis increases greatly during the tomato ripening process due to the conversion of chloroplasts in chromoplasts, taking the place of chlorophyll [20, 30]. In addition to the chromatic indices, TSS or TA values were related to the ripening process of fruits, strongly affecting sweetness, sourness and flavor of fresh and processed horticultural products [31]. The TSS value indicates dissolved amounts of solids (sugars and organic acids, together with small amounts of vitamins, proteins, pigments, phenolics and minerals) present in a food [32]. Since sugars (glucose and fructose) constitute most of the total soluble solids in a tomato fruit (approximately 65–70%) [30], TSS content for this category of food could be associated in good approximation with sugar content and, consequently, with fruit sweetness.

In this study, statistical analysis of the colorimetric data highlighted that non-PGI tomatoes (Lots 1 and 2) were redder, and riper, than PGI tomatoes (Lots 3 and 4), whereas chemical parameters showed similarity, especially in solid soluble (TSS) content, between Lots 1 and 4 from one side and Lots 2 and 3 on the other side. Thus, sweetness cannot be associated to the

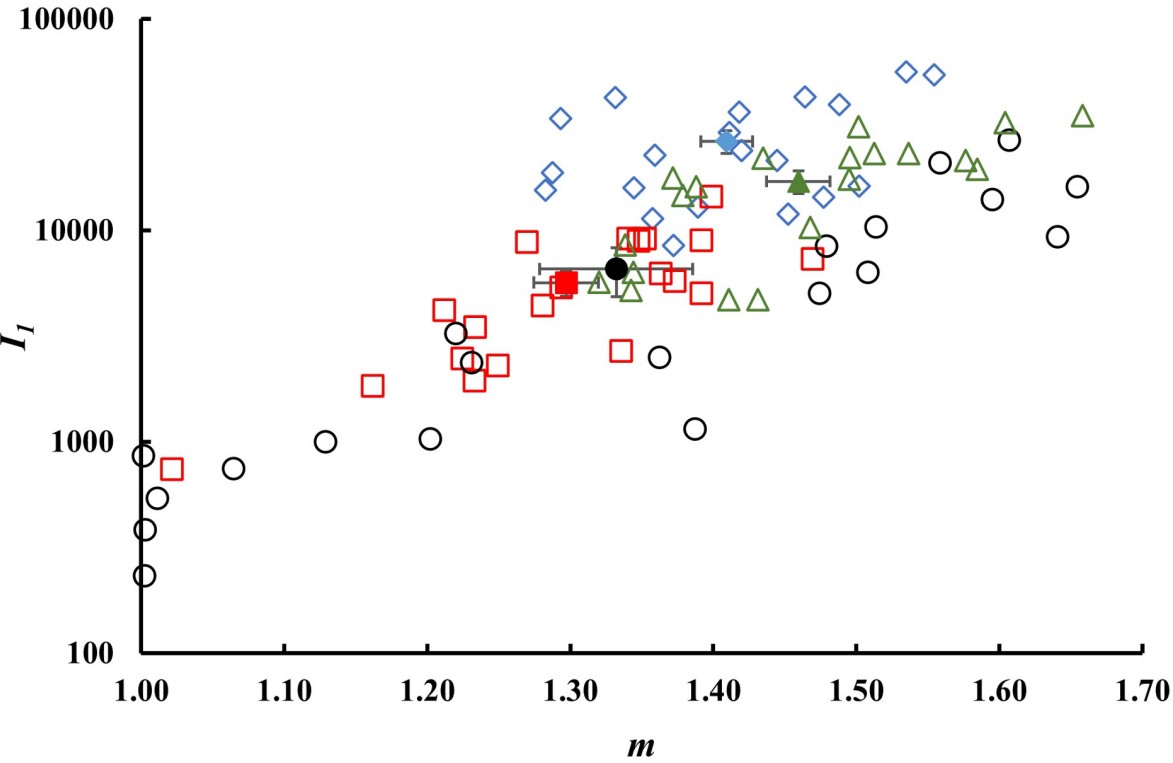

**Fig 5. Comparison between experimental data and decay slope.** First experimental point $I_1$ of DL decay as a function of the parameter $m$ obtained through the fitting procedure according to Eq (1) of the DL normalised decay curves in correspondence of each fruit which was part of a Lot: (diamond) Lot 1, (square) Lot 2, (circle) Lot 3, (triangle) Lot 4. Errors of $m$ parameter are within the marker size. Poissonian errors (square roots) of the count $I_1$ are omitted for the sake of clarity. Full markers represent mean values ±. S.E. for each Lot.

geographical origin or to agricultural practices distinguishing PGI tomatoes from non-PGI ones.

The physicochemical transformations induced in biological systems by biotic and abiotic factors can be brought to light also through non-destructive physical techniques. A biological system suitably irradiated with a source of electromagnetic radiation in different spectral bands will generate in response different signals in relation to its qualitative profile, defined by the content of chemical elements and organic compounds present in it. Spectroscopic methods involving electromagnetic radiations, such as Raman, NIRs or XRF spectroscopy are sensitive to changes involving food chemical composition. Thus, they are widely used to discriminate foodstuffs on the basis of their geographical origin or cultivation techniques [33–35] or to detect the presence of chemical pesticides and toxic substances, such as mycotoxins [5, 36, 37].

In this study, we tested the possibility to apply DL technique as a tool to discriminate tomato Lots of the same cultivar subjected to different agronomic practices and farming conditions (PGI and non-PGI tomato samples) based on some parameters linked to the sensory quality of ripe fruits. In this regard, the possibility of testing and quantifying differences between vegetable crops through non-destructive methods are of great industrial relevance.

The comparisons made on $t_0$ and $m$ parameters associated to decay curves of PGI and non-PGI tomatoes highlighted significant differences between the four Lots analyzed. As previously observed for TSS measurements, the slopes of curves reached the highest values for Lots 1 and 4. Similarly, the values associated with the parameter $t_o$ and experimental first point $I_1$ differed in the four tomato Lots analyzed. Overall, it was observed that Lots from the same growing

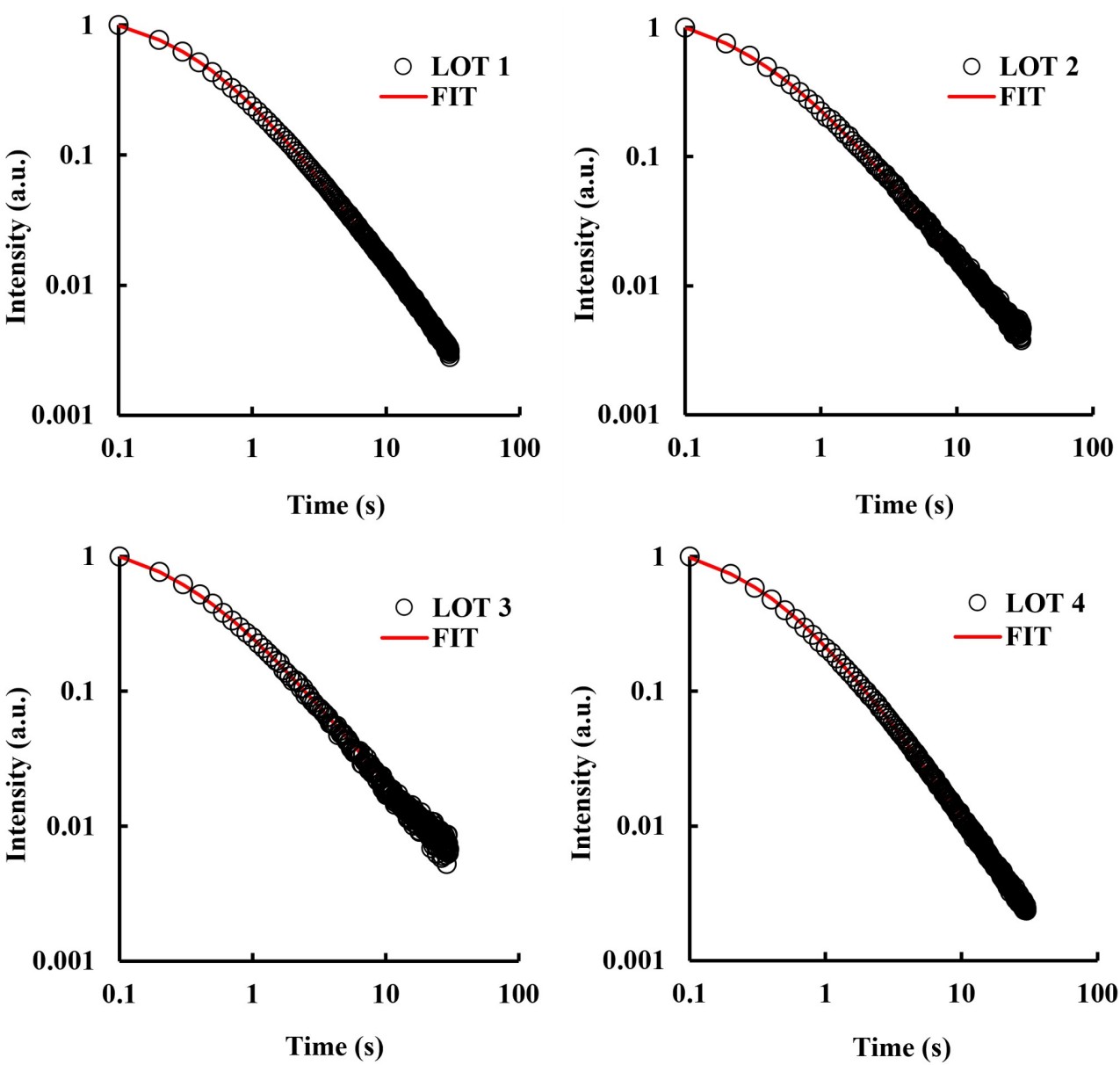

**Fig 6. Averaged normalized decay trend.** For each tomato Lot under study, the average normalized decay trend was evaluated (see text). Solid lines refer to the theoretical trend according to Eq (1). Data are plotted on a log-log scale.

area exhibited different DL decay curves, while Lots from different growing areas had similar temporal trends. Thus, the analysis of DL data set actually does not support a classification by production areas and, consequently, it is not suitable for authenticating vegetal foodstuffs. This should not be surprising, if we consider the kind of information that can be retrieved by this technique. Unlike other optical spectroscopies, Delayed Luminescence is hardly applicable to quantitative compositional analyses or Lot authentication, as the counted photons cannot be directly related to characterizing chemical components. As said, DL is an indicator of the functional state of a biological system, as it highlights structural and/or functional changes in cells induced by certain metabolic processes [38, 39]. External agents that affect plant

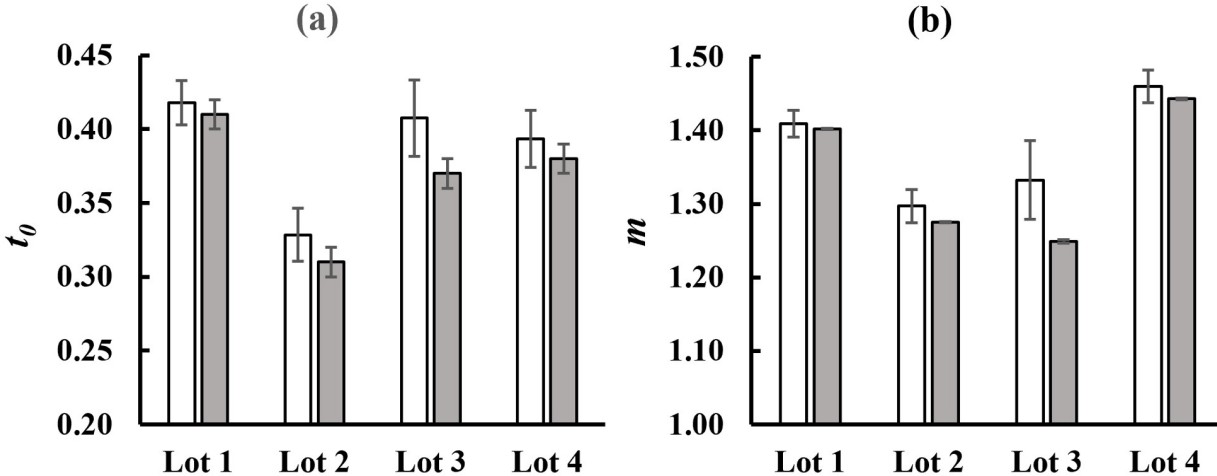

**Fig 7. Comparison between DL parameters obtained by using two different procedures.** DL parameters obtained according to Eq (1) are reported for each Lot: (white columns) mean values of the DL parameters obtained by fitting the single fruit DL decay trends, (grey columns) DL parameters obtained by fitting the averaged normalized decay trend. Bars indicate the standard errors. (a) $t_0$ parameter, (b) $m$ parameter.

photosynthesis or chloroplast content (physiological stresses, atmospheric pollutants, agricultural practices, etc.) often induce structural changes in biological systems that in turn lead to changes in the number of optical photons released [3]. In this regard, recent studies have shown that biological organisms and herbal medicines can release different light signals as growing environments, agronomic practices and technological processes vary [22, 40–42]. However, based on our preliminary results, we believe that it is difficult to correlate any structural changes to the territory of origin, especially in the case of ripe fruit growing under similar climatic conditions, as numerous variable can arise inducing different DL signals.

To assess whether luminescence can be used as an indicator of quality, DL parameters were correlated with the chemical parameters previously determined for each Lot. Hence, to perform a correlation analysis, we considered the slopes of the DL curves ($m$ parameter), whose average values showed the largest variability between Lots (see Fig 7b). In particular, the correlations between the $m$ parameter and TSS, TA or MI were studied by considering the mean values of the parameters on the entire set of data. As previously reported, sugars, acids and their ratio are important to determine sweetness, sourness and overall flavor intensity [31]. Tomato fruits with a high level of acids and sugars have good flavor, while those with a low level of acids or sugars result in insipid or tart tomatoes, respectively. The TSS, TA and TSS/TA parameters are also used to determine the best time for harvesting, since they change considerably during fruit ripening. The TSS content tends to increase as a result of biosynthesis of sugars, while TA increases during tomato development, reaching a maximum at the breaker stage, and then it decreases as ripening progresses [43].

The results, shown in Fig 8, revealed linear relationship ($R^2 = 0.994$) between the fit slopes ($m$ values) and measured TSS parameters. Poor correlation ($R^2 = 0.49$) or no correlation ($R^2 = 0.29$) was found between $m$ and maturity index or citric acid content, respectively (S1 and S2 Figs). Interestingly, a still good relationship ($R^2 = 0.976$) holds if we use the $m$-values related to the averaged normalized decay trends of Fig 6 instead of the average $m$-parameters.

The phenomenological connection between quality and photon emission rate (to which the $m$ parameter is correlated) has been found for other samples and is reported in literature. In this regard, some studies have found correlations between DL signals and color of tomato fruits at different maturity stages [14], as well as between DL intensity and fungal

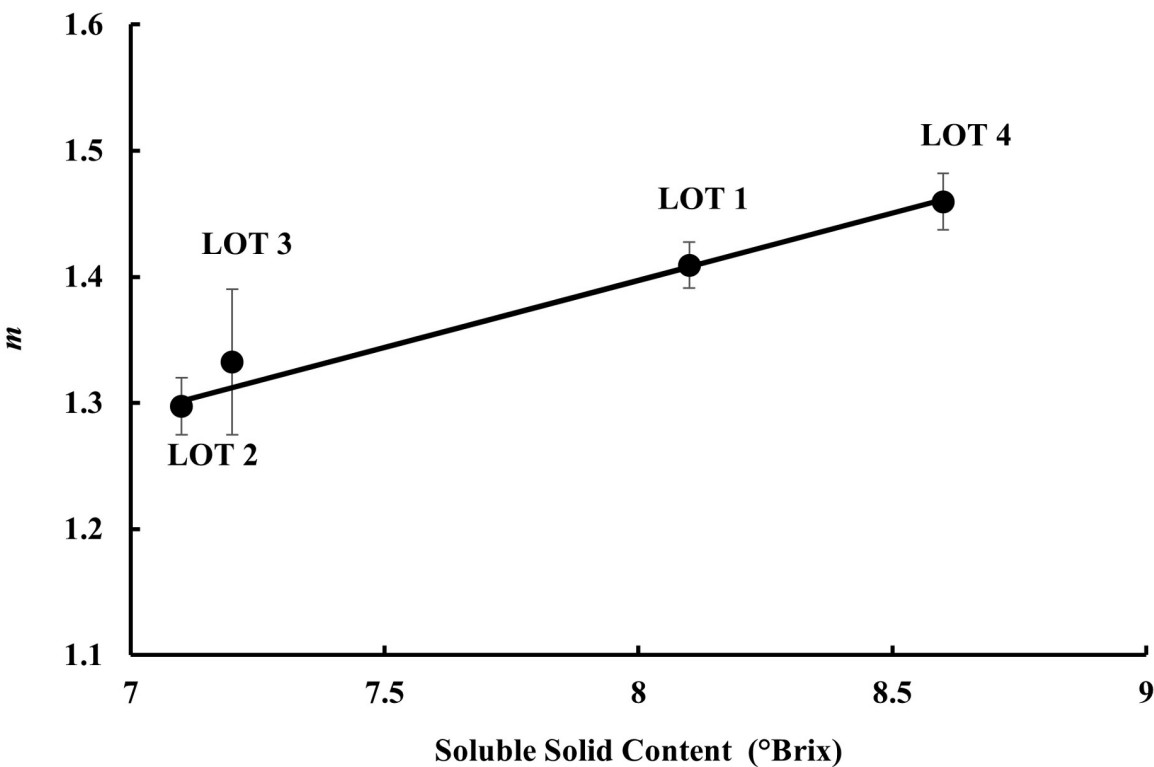

**Fig 8. Comparison between chemical and DL features.** DL decay slope (m value) as a function of the total soluble solid content (TSS) is reported. Markers represent the average values obtained, for each Lot, from the single fruit DL trends. Bars denotes standard errors. Solid line refers to the weighted linear fit of experimental data, according to the equation y = (0.11 ± 0.01) x + (0.54 ± 0.05), with $R^2$ = 0.994.

contamination level in peanuts artificially inoculated with *A. flavus* [44]. As previously reported, DL measurements were able to differentiate medicinal herbs prepared under different conditions and showing different age and therapeutic properties (these characteristics reflect the presence of different active chemical compounds) [21, 22, 40]. Our preliminary results are in agreement with findings reported in literature and suggest a possible use of DL technique as an indicator of fruit internal quality.

Finally, our results showed negligible differences between DL parameters obtained from the two fitting procedures explained in paragraph 3.2. These differences were less than 5% for most samples under study (see Fig 7). Thus, it can be considered that the fitting procedure applied on the average decay trends, obtained by averaging full dataset associated with each Lot, is a valid tool to be used for fast analysis of a large amount of data.

## 5. Conclusions

The feasibility of DL technique in determining relevant quality features affecting the flavor, commerciality and maturity degree of vegetal crops was investigated on cherry tomato samples harvested at full ripeness stage (fully red skin) from two different growing areas of south-eastern Sicily. In addition, we have also explored the possibility of applying DL technique to disentangle tomato Lots on the basis of their geographical origins. The tomato fruits under study appeared similar in colour, pH and acid content, while differing in the total soluble solid content.

The preliminary results achieved in this study suggest that DL is not suitable for authenticating vegetal foodstuffs, as the changes in the DL parameters in relation to the different DL decay curves ($m$ and $t_0$ values) were not correlated to the territorial origin of the Lots. In our opinion, DL cannot discriminate similar products, that differ from each other for few chemical components [1, 2]. As a matter of fact, DL highlights the functional properties of biological systems when they are subjected to significant structural changes.

Conversely, the differences in photon decay rate emitted by the tomato fruits after their illumination reflected the differences in total soluble solids content present in their juice. Thus, we believe that DL technique has a good potential, being a rapid technique which can help farmers understand, in real time, the quality of their products. The assessment of the flavor and maturity stage of tomato fruits through the DL technique, in addition or in place of the traditional chemical methodologies, would offer in the agri-food sector many advantages. In particular, it would allow a fast quality assessment without damaging or destroying the analyzed biological matrices, thus providing a large number of measurements and more accurate results in a short time.

## Supporting information

**S1 Appendix. DL datasets representing the number of counts released by tomato fruits in a time interval of 30 s after excitation.**
(XLSX)

**S1 Fig. DL decay slope (m value) as a function of Maturity Index (TSS/TA).**
(PDF)

**S2 Fig. DL decay slope (m value) as a function of citric acid content (TA).**
(PDF)

## Acknowledgments

The authors are thankful to Dr. Giuseppe Brafa, head of the Colle D'oro Agricultural Company in Ispica, and to all the agricultural entrepreneurs who provided with the tomato samples being studied, kindly hosting us on their greenhouses.

## Author Contributions

**Conceptualization:** Salvina Panebianco, Gabriella Cirvilleri, Agatino Musumarra, Maria Grazia Pellegriti, Agata Scordino.

**Formal analysis:** Salvina Panebianco, Gabriella Cirvilleri, Alberto Continella, Maria Grazia Pellegriti, Agata Scordino.

**Funding acquisition:** Gabriella Cirvilleri, Agata Scordino.

**Investigation:** Salvina Panebianco, Yu Yan, Giulia Modica.

**Methodology:** Salvina Panebianco, Eduard van Wijk, Yu Yan, Gabriella Cirvilleri, Alberto Continella, Giulia Modica, Agatino Musumarra, Maria Grazia Pellegriti, Agata Scordino.

**Project administration:** Gabriella Cirvilleri, Agatino Musumarra.

**Resources:** Salvina Panebianco, Eduard van Wijk, Yu Yan, Gabriella Cirvilleri, Alberto Continella, Giulia Modica, Agatino Musumarra, Maria Grazia Pellegriti, Agata Scordino.

**Supervision:** Eduard van Wijk, Gabriella Cirvilleri, Agatino Musumarra, Agata Scordino.

**Validation:** Salvina Panebianco, Eduard van Wijk, Yu Yan, Gabriella Cirvilleri, Alberto Continella, Giulia Modica, Maria Grazia Pellegriti, Agata Scordino.

**Visualization:** Salvina Panebianco, Agata Scordino.

**Writing – original draft:** Salvina Panebianco, Agata Scordino.

**Writing – review & editing:** Salvina Panebianco, Eduard van Wijk, Yu Yan, Gabriella Cirvilleri, Alberto Continella, Giulia Modica, Agatino Musumarra, Maria Grazia Pellegriti, Agata Scordino.

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
