## [Decision Letter · Decision Letter 0]

2 May 2023

PONE-D-23-08412Application of Delayed Luminescence for assessing quality of tomato fruit coming from different Sicilian growing areasPLOS ONE

Dear Dr. Panebianco,

Thank you for submitting your manuscript to PLOS ONE. After careful consideration, we feel that it has merit but does not fully meet PLOS ONE’s publication criteria as it currently stands. Therefore, we invite you to submit a revised version of the manuscript that addresses the points raised during the review process.

Please see below, as well as attached file

We look forward to receiving your revised manuscript.

Kind regards,

Charles Odilichukwu R. Okpala

Academic Editor

PLOS ONE

Journal Requirements:

"Funded by Incentive Plan for Research (PIA.CE.RI.) 2020-22 line 2 of the University of Catania, Research Projects “NaTI4Smart” and “MEDIT ECO”, National Operational Programme (PON) on Research and Innovation 2014-2020 “POFACS (ARS01_00640)” - Italian Ministry of University and Research. The funders had no role in study design, data collection and analysis, decision to publish, or preparation of the manuscript.

We note that one or more of the authors is affiliated with the funding organization, indicating the funder may have had some role in the design, data collection, analysis or preparation of your manuscript for publication; in other words, the funder played an indirect role through the participation of the co-authors. If the funding organization did not play a role in the study design, data collection and analysis, decision to publish, or preparation of the manuscript and only provided financial support in the form of authors' salaries and/or research materials, please do the following:

(1) Review your statements relating to the author contributions, and ensure you have specifically and accurately indicated the role(s) that these authors had in your study. These amendments should be made in the online form.

(2) Confirm in your cover letter that you agree with the following statement, and we will change the online submission form on your behalf: 

“The funder provided support in the form of salaries for authors [insert relevant initials], but did not have any additional role in the study design, data collection and analysis, decision to publish, or preparation of the manuscript. The specific roles of these authors are articulated in the ‘author contributions’ section.

6. Please amend either the title on the online submission form (via Edit Submission) or the title in the manuscript so that they are identical.

7. We note that Figure 1 in your submission contain map images which may be copyrighted. All PLOS content is published under the Creative Commons Attribution License (CC BY 4.0), which means that the manuscript, images, and Supporting Information files will be freely available online, and any third party is permitted to access, download, copy, distribute, and use these materials in any way, even commercially, with proper attribution. For these reasons, we cannot publish previously copyrighted maps or satellite images created using proprietary data, such as Google software (Google Maps, Street View, and Earth). For more information, see our copyright guidelines: http://journals.plos.org/plosone/s/licenses-and-copyright.

(1) You may seek permission from the original copyright holder of Figure 1 to publish the content specifically under the CC BY 4.0 license.  

**Additional Editor Comments:**

Please attend to the comments raised by the reviewers.

Reviewers' comments:

Reviewer's Responses to Questions

**Comments to the Author**

1. Is the manuscript technically sound, and do the data support the conclusions?

Reviewer #1: Yes

Reviewer #2: Yes

2. Has the statistical analysis been performed appropriately and rigorously? 

Reviewer #1: Yes

Reviewer #2: Yes

3. Have the authors made all data underlying the findings in their manuscript fully available?

Reviewer #1: Yes

Reviewer #2: Yes

4. Is the manuscript presented in an intelligible fashion and written in standard English?

Reviewer #1: Yes

Reviewer #2: Yes

5. Review Comments to the Author

Reviewer #1: 1) Line 118 should be physicochemical

2) If statistical letters are written as superscripts will be much easy to understand.

3) In the paper says it is difficult to apply DL for all produced tomatoes at least on a farm. So how can the method be applicable?

Reviewer #2: Generally well written, however, some revisions have been suggested. Generally well written, however, some revisions have been suggested. Please see attachments. Generally well written, however, some revisions have been suggested. Please see attachments.

6. PLOS authors have the option to publish the peer review history of their article (what does this mean?). If published, this will include your full peer review and any attached files.

Reviewer #1: No

Reviewer #2: **Yes: **Erastus Mak-Mensah

---

## [Author Response · Author response to Decision Letter 0]

12 May 2023

Reviewer #1: 

Q1) Line 118 should be physicochemical.

A1) According to the suggestion, “physic-chemical” has been replaced by “physicochemical” in the line 118 of the manuscript and in various other parts of text.

Q2) If statistical letters are written as superscripts will be much easy to understand.

A2) Concerning this suggestion, we wrote the statistical letters as superscripts, both in Table 1 and in Table 2. 

Q3) In the paper says it is difficult to apply DL for all produced tomatoes at least on a farm. So how can the method be applicable?

A3) Actually, we do not understand this question. Nowhere in the paper does it say “it is difficult to apply DL for all produced tomatoes at least on a farm”, neither such a conclusion can be drawn by our experimental study. Conversely, our experimental study supports the idea that, after standardization, a no-destructive method based on Delayed Luminescence to check single fruit quality could be implemented.

Reviewer #2: 

Generally well written, however, some revisions have been suggested. Please see attachments.

A) We have revised the manuscript, according to the comments suggested in attachments. As requested, we have also rewritten some sentences to clarify our concepts in lines 79-84, 87-91, 230-236, 371-375, 385-389 and 403-408 of the manuscript.

---

## [Decision Letter · Decision Letter 1]

16 May 2023

Applications of Delayed Luminescence for tomato fruit quality assessment across varied Sicilian cultivation zones

PONE-D-23-08412R1

Dear Dr. Panebianco,

We’re pleased to inform you that your manuscript has been judged scientifically suitable for publication and will be formally accepted for publication once it meets all outstanding technical requirements.

Kind regards,

Charles Odilichukwu R. Okpala

Academic Editor

PLOS ONE

Additional Editor Comments (optional):

The current revised manuscript is acceptable for publication.

Reviewers' comments:

Reviewer's Responses to Questions

**Comments to the Author**

1. If the authors have adequately addressed your comments raised in a previous round of review and you feel that this manuscript is now acceptable for publication, you may indicate that here to bypass the “Comments to the Author” section, enter your conflict of interest statement in the “Confidential to Editor” section, and submit your "Accept" recommendation.

Reviewer #1: All comments have been addressed

Reviewer #2: All comments have been addressed

2. Is the manuscript technically sound, and do the data support the conclusions?

Reviewer #1: Yes

Reviewer #2: Yes

3. Has the statistical analysis been performed appropriately and rigorously? 

Reviewer #1: Yes

Reviewer #2: Yes

4. Have the authors made all data underlying the findings in their manuscript fully available?

Reviewer #1: Yes

Reviewer #2: Yes

5. Is the manuscript presented in an intelligible fashion and written in standard English?

Reviewer #1: Yes

Reviewer #2: Yes

6. Review Comments to the Author

Reviewer #1: There is no comment to the authors for research, dual, or publication ethics. There is no additional comments.

Reviewer #2: Dear Authors, after carefully examining your revised manuscript, I am happy to recommend for acceptance and further publication. The manuscript is in good order. Keep up the good work.

7. PLOS authors have the option to publish the peer review history of their article (what does this mean?). If published, this will include your full peer review and any attached files.

Reviewer #1: No

Reviewer #2: **Yes: **Erastus Mak-Mensah

---

## [Editor Report · Acceptance letter]

22 May 2023

PONE-D-23-08412R1 

Applications of Delayed Luminescence for tomato fruit quality assessment across varied Sicilian cultivation zones 

Dear Dr. Panebianco:

I'm pleased to inform you that your manuscript has been deemed suitable for publication in PLOS ONE. Congratulations! Your manuscript is now with our production department. 

Kind regards, 

on behalf of

Dr. Charles Odilichukwu R. Okpala 

Academic Editor

PLOS ONE